computational chemistry/medicinal chemistry

sulfuretin, DFT study, antioxidants, antiradical activity, flavonoids

**Author for correspondence:**
Quan V. Vo
e-mail: vvquan@ute.udn.vn

This article has been edited by the Royal Society of Chemistry, including the commissioning, peer review process and editorial aspects up to the point of acceptance.

# The hydroperoxyl radical scavenging activity of sulfuretin: insights from theory

Nguyen Thi Hoa[1], Do Thi Ngoc Hang[1], Do Phu Hieu[1], Huynh Van Truong[1], Loc Phuoc Hoang[2], Adam Mechler[3] and Quan V. Vo[1]

[1]The University of Danang – University of Technology and Education, Danang 550000, Vietnam
[2]Quang Tri Teacher Training College, Quang Tri province 520000, Vietnam
[3]Department of Chemistry and Physics, La Trobe University, Victoria 3086, Australia

QVV, 0000-0001-7189-9584

Sulfuretin (SFR), which is isolated from *Rhus verniciflua, Toxicodendron vernicifluum, Dahlia, Bidens tripartite* and *Dipterx lacunifera*, is one of the most important natural flavonoids. This compound is known to have numerous biological activities; among these, the antioxidant activity has not been thoroughly studied yet. In this study, the hydroperoxyl scavenging activity of SFR was examined by using density functional theory calculations. SFR is predicted to be an excellent HOO$^{\bullet}$ scavenger in water at pH = 7.40 with $k_{overall} = 4.75 \times 10^7$ M$^{-1}$ s$^{-1}$, principally due to an increase in the activity of the anionic form following the single-electron transfer mechanism. Consistently, the activity of the neutral form is more prominent in the non-polar environment with $k_{overall} = 1.79 \times 10^4$ M$^{-1}$ s$^{-1}$ following the formal hydrogen transfer mechanism. Thus, it is predicted that SFR exhibits better HOO$^{\bullet}$ antiradical activity than typical antioxidants such as resveratrol, ascorbic acid or Trolox in the lipid medium. The hydroperoxyl radical scavenging of SFR in the aqueous solution is approximately 530 times faster than that of Trolox and similar to ascorbic acid or resveratrol. This suggests that SFR is a promising radical scavenger in physiological environments.

## 1. Introduction

Sulfuretin (SFR, figure 1) is a natural flavonoid present in numerous plant species, including *Rhus verniciflua* [1,2], *Toxicodendron vernicifluum* [3], *Dahlia, Bidens tripartite* and *Dipterx lacunifera* [4]. This compound is known to have numerous biological activities

Sulfuretin

**Figure 1.** Molecular structure and atomic numbering of SFR.

such as amelioration of rheumatoid arthritis symptoms [5], antimutagenic [6], antiplatelet [7], anti-cancer [8,9], anti-inflammatory effects [5,10], liver protection [11], anti-ageing effect for skin [12], anti-obesity effect [12] and antioxidant activity [2,13–15].

Jung and co-workers [2] reported that SFR presented strong antioxidant activity in the DPPH (2,2-diphenyl-1-picrylhydrazyl) assay and total anti-ROS (reactive oxygen species) activity with IC50 = 8.52 and 0.73 µM, respectively. The DPPH inhibition of SFR was about two times higher than that of L-ascorbic acid, whereas the total ROS inhibition is about five times stronger than Trolox. SFR also presented significant activity against $ONOO^-$ and $HO^•$ radicals [2]. Chen *et al.* [14] also reported that SFR has good DPPH, $ABTS^{•+}$ (2,2′-azino-bis(3-ethylbenzothiazoline-6-sulfonic acid) and $HO^•$ radical scavenging activity that is higher than butylated hydroxytoluene (BHT).

Although the antioxidant activity of SFR is broadly examined experimentally [2,14], there are no studies on the mechanism and kinetics of its antiradical activity, particularly in physiological environments. Computer calculations offer a convenient way to predict the antioxidant activity of organic compounds in physiological media [16–23]. In this context and as a continuation of our previous studies [18,24,25], we set out in this work to evaluate the $HOO^•$ antiradical activity of SFR by a combination of thermodynamic and kinetic calculations. This study also considered the effects of solvents on the antioxidant properties of SFR in comparison with some typical antioxidants.

# 2. Computational details

All calculations were carried out with Gaussian 09 suite of programs [26]. M06–2X/6–311 + +G(d,p) model chemistry was used for all calculations [27–29]. It was demonstrated before that the M06–2X functional is one of the most reliable methods to study thermodynamics and kinetics of radical reactions, particularly in physiological environments [19,28,30,31]. The solvation model density (SMD) method was used for including the effects of water and pentyl ethanoate in the computations [17,18,24,32–34]. The kinetic calculations were performed following the quantum mechanics-based test for the overall free radical scavenging activity (QM-ORSA) protocol [17,34], using the conventional transition state theory (TST) and 1 M standard state at 298.15 K [18,34–40]. The details of the method are shown in the electronic supplementary material, table S1.

# 3. Results and discussion

## 3.1. The HOO˙ antiradical activity of SFR in the gas phase

### 3.1.1. Thermodynamic evaluation

For SFR that contains OH and moieties, the antioxidant activity may follow either of the four main mechanisms: the formal hydrogen transfer (FHT), the sequential proton loss electron transfer (SPLET), the single-electron transfer proton transfer (SETPT) and radical adduct formation (RAF) [41,42]. The first three pathways are defined by the following thermodynamic parameters: bond dissociation enthalpy (BDE), proton affinity (PA) and ionization energy (IE), respectively. The Gibbs free energy change of the addition reaction is calculated directly for the RAF mechanism. Thus, the BDE, IE and PA values of SFR were first calculated in the gas phase, and the results are shown in table 1.

As per table 1, the lowest BDE value was predicted for O4′−H at 77.5 kcal mol$^{-1}$. This value is lower than that of natural antioxidants such as viniferifuran (82.7 kcal mol$^{-1}$) [43], resveratrol (83.9 kcal mol$^{-1}$)

**Table 1.** The calculated thermodynamic parameters (BDEs, PAs and IEs) of SFR in the gas phase.

| positions | BDE | PA | IE |
|---|---|---|---|
| O6—H | 90.7 | 323.4 | 174.6 |
| O3′—H | 80.5 | 327.9 | |
| O4′—H | 77.5 | 320.9 | |

**Table 2.** Calculated $\Delta G^o$ (kcal/mol) of the SFR + HOO$^\bullet$ reactions according to the FHT, SP, RAF and SET mechanisms in the gas phase.

| positions | FHT | SP | SET | RAF |
|---|---|---|---|---|
| O6—H | 4.8 | 170.8 | 152.1 | — |
| O3′—H | −4.9 | 176.1 | | — |
| O4′—H | −7.7 | 169.2 | | — |
| C2 | — | — | | 1.1 |
| C8 | — | — | | −4.6 |

[43], puerarin (87.3 kcal mol$^{-1}$) [44] and vanillic acid (85.2 kcal mol$^{-1}$) [45]. The lowest PA and IE values are about 4.14 and 2.25 times higher than the BDE value. Thus, based on the computed data, the antioxidant activity of SFR is predicted to favour the FHT pathway, at least in apolar and low-dielectric environments.

To confirm that FHT is indeed the preferred pathway of the HOO$^\bullet$ antiradical activity of SFR, the Gibbs free energy of the SFR + HOO$^\bullet$ reaction was calculated according to each of the four mechanisms: FHT, single-electron transfer (SET, the first step of the SETPT mechanism), sequential proton (SP, the first step of the SPLET) and RAF (table 2). It was found that the HOO$^\bullet$ antiradical activity of SFR is only clearly spontaneous for FHT at O3′(O4′)—H bonds and RAF at the C8 position ($\Delta G^o < 0$), whereas the RAF reaction at C2 with $\Delta G^o = 1.1$ kcal mol$^{-1}$ cannot be clearly excluded based on thermodynamics alone and therefore it was also included in the kinetic study. The other reactions are clearly not spontaneous with high positive $\Delta G^o$ values. The $\Delta G^o$ values for the reactions following the SP and SET pathways are much higher than those of the FHT mechanism. Thus, the calculated data suggest that the HOO$^\bullet$ antiradical activity of SFR may follow either FHT or RAF mechanism (at O3′(4′)—H and C2/C8 positions, respectively), and these pathways should be investigated in the kinetic study.

### 3.1.2. Kinetic study

Based on the above results, the kinetics of the SFR + HOO$^\bullet$ reaction in the gas phase was investigated for the thermodynamically favourable positions and mechanisms according to the QM-ORSA protocol [17], and the data are presented in table 3 and figure 2.

It is apparent that the HOO$^\bullet$ antiradical activity of SFR occurs mostly by the H-abstraction of the O4′—H bond ($\Delta G^{\neq} = 11.2$ kcal/mol; $k_{Eck} = 2.83 \times 10^6$ M$^{-1}$ s$^{-1}$; $\Gamma = 77.0\%$). That is more than three times higher contribution than the hydrogen abstraction of the O3′—H bond ($\Delta G^{\neq} = 11.6$ kcal mol$^{-1}$; $k_{Eck} = 8.43 \times 10^5$ M$^{-1}$ s$^{-1}$; $\Gamma = 23.0\%$). By contrast, the addition of the radical does not make any contribution ($\Gamma = 0\%$) at either the C2 or C8 positions. This result is in good agreement with previous studies in phenolic compounds [46–48]. We can conclude that the HOO$^\bullet$ antiradical activity of SFR is dominated by the FHT mechanism at the O3′(4′)—H bond; therefore, this is further analysed in physiological environments.

## 3.2. The HOO$^\bullet$ antiradical activity of SFR in physiological environments

### 3.2.1. Acid–base equilibrium

Previous studies showed that the deprotonation of the OH bonds plays a key role in the HOO$^\bullet$ antiradical activity of phenolic compounds in the aqueous solution [30,34,49]. The spontaneous dissociation of acidic moieties practically eliminates the activation energy barrier of the first step of the SPLET mechanism, simplifying it to direct electron transfer, and for this reason, this pathway can become energetically favoured in aqueous solution for the dissociated species. Thus, in this study, the

**Figure 2.** The optimized transition state (TS) structures following the FHT and RAF mechanisms of the SFR + HOO⁺ reaction (G: gas phase; W: water; P: pentyl ethanoate).

**Figure 3.** The acid dissociation equilibrium of SFR.

**Table 3.** Calculated $\Delta H$ (kcal/mol), activation Gibbs free energies ($\Delta G^{\neq}$, kcal/mol), tunnelling corrections ($\kappa$), $k_{Eck}$ (M$^{-1}$ s$^{-1}$) and branching ratios ($\Gamma$, %) for the HOO⁺ + SFR reaction in the gas phase.

| mechanism | positions | $\Delta H$ | $\Delta G^{\neq}$ | $\kappa$ | $k_{Eck}$ | $\Gamma$ |
|---|---|---|---|---|---|---|
| FHT | O3'−H | 2.3 | 11.6 | 39.6 | $8.43 \times 10^5$ | 23.0 |
|  | O4'−H | 2.0 | 11.2 | 72.1 | $2.83 \times 10^6$ | 77.0 |
| RAF | C2 | 7.1 | 17.1 | 1.5 | 2.83 | 0.0 |
|  | C8 | 8.6 | 17.7 | 1.5 | $9.03 \times 10^{-1}$ | 0.0 |
| $k_{overall}$ |  |  |  |  | $3.67 \times 10^6$ |  |

deprotonation of SFR must also be considered. The PA values (table 1) showed that the site most likely to dissociate is the O4'−H bond. Thus, this bond was used to calculate the pKa of SFR. The pKa was computed following the literature [49,50], and the results are shown in figure 3. The calculated pKa value was 7.47. Thus, under physiologically relevant conditions (pH = 7.40), SFR has both neutral (HA, 54.0%) and anionic (A⁻, 46.0%) forms. Therefore, in the physiological environments, these states were used for the kinetic investigation.

**Table 4.** Calculated $\Delta G^{\#}$ (kcal mol$^{-1}$), tunnelling corrections ($\kappa$), the nuclear reorganization energy ($\lambda$, kcal mol$^{-1}$) rate constants ($k_{app}$, $k_f$ and $k_{overall}$ M$^{-1}$ s$^{-1}$), molar fractions ($f$) and branching ratios ($\Gamma$, %) at 298.15 K, in the SFR + HOO· reaction in pentyl ethanoate and water solvents.

| mechanism | | pentyl ethanoate | | | | water | | | | | |
|---|---|---|---|---|---|---|---|---|---|---|---|
| | | $\Delta G^{\#}$ | $\kappa$ | $k_{app}$ | $\Gamma$ | $\Delta G^{\#}$ | $\kappa$ | $k_{app}$ | $f$ | $k_f^{**}$ | $\Gamma$ |
| SET | | | | | | 6.6 | 15.6* | $8.90 \times 10^7$ | 0.460 | $4.09 \times 10^7$ | 86.2 |
| HAT | O3'–H | 15.0 | 106.9 | $6.90 \times 10^3$ | 38.5 | 16.0 | 744.5 | $9.20 \times 10^3$ | 0.540 | $4.97 \times 10^3$ | 0.0 |
| | O4'–H | 14.9 | 163.1 | $1.10 \times 10^4$ | 61.5 | 15.5 | 202.8 | $5.30 \times 10^3$ | 0.540 | $2.86 \times 10^3$ | 0.0 |
| | O3'–H (anion) | | | | | 7.8 | 1.2 | $1.42 \times 10^7$ | 0.460 | $6.53 \times 10^6$ | 13.7 |
| $k_{overall}$ | | | | $1.79 \times 10^4$ | | | | | | $4.75 \times 10^7$ | |

*$\lambda$; **$k_f = f.k_{app}$; $\Gamma = k.100/k_{overall}$.

### 3.2.2. Kinetic study

Based on the results of the kinetic calculations in the gas phase, the HOO$^\bullet$ antiradical activity in non-polar environments was modelled by the hydrogen transfer mechanism at the O3′(O4′)–H bonds. In the aqueous environment, the SET mechanism was also investigated for the deprotonated state of SFR. The overall rate constants ($k_{overall}$) were computed following the QM-ORSA protocol [17,33], (table 4) according to equations (3.1) and (3.2).

In the lipid medium

$$k_{overall} = \Sigma k_{app}(\text{FHT}(O-H) - \text{neutral}). \tag{3.1}$$

In water at pH = 7.40

$$k_{overall} = \Sigma k_f(\text{FHT} - \text{neutral}) + k_f(\text{SET} - \text{anion}) + k_f(\text{FHT}(O3' - H) - \text{anion}). \tag{3.2}$$

As shown in table 4, the HOO$^\bullet$ antiradical activity of SFR in the polar solvent is excellent with the $k_{overall} = 4.75 \times 10^7 \, \text{M}^{-1} \, \text{s}^{-1}$. Similarly, in the lipid medium, SFR exhibits good activity with $k_{overall} = 1.79 \times 10^4 \, \text{M}^{-1} \, \text{s}^{-1}$. It was found that the SET of anion A$^-$ plays a principal role ($k_f = 4.09 \times 10^7 \, \text{M}^{-1} \, \text{s}^{-1}$, $\Gamma = 86.2\%$) in the radical scavenging activity of SFR. The H-abstraction of the anion state contributes about 13.7% to the overall rate constants. The rate constants for the H-abstraction of O3′(O4′)–H bonds against HOO$^\bullet$ radical are $k_f = 4.97 \times 10^3$ and $2.86 \times 10^3 \, \text{M}^{-1} \, \text{s}^{-1}$, respectively; however, these reactions do not make any contributions (approx. 0%) to the activity of SFR. Based on the results, SFR is better HOO$^\bullet$ radical scavenger than typical antioxidants Trolox, ascorbic acid and resveratrol in lipid phase (reference lipid phase activities: $k_{overall} = 3.40 \times 10^3 \, \text{M}^{-1} \, \text{s}^{-1}$ [33], $k_{overall} = 5.71 \times 10^3 \, \text{M}^{-1} \, \text{s}^{-1}$ [17] and $k_{overall} = 1.31 \times 10^4 \, \text{M}^{-1} \, \text{s}^{-1}$ [46], respectively). In aqueous solution, the HOO$^\bullet$ antiradical activity of SFR is approximately 530 times faster than that of Trolox ($k = 8.96 \times 10^4 \, \text{M}^{-1} \, \text{s}^{-1}$) [33] and fairly similar to other well-known natural antioxidants, i.e. ascorbic acid ($k = 9.97 \times 10^7 \, \text{M}^{-1} \, \text{s}^{-1}$) [17] and resveratrol ($k = 5.62 \times 10^7 \, \text{M}^{-1} \, \text{s}^{-1}$) [46]. Thus, the results suggest that SFR is a promising antioxidant in physiological media.

## 4. Conclusion

The hydroperoxyl radical scavenging activity of SFR was investigated using DFT calculations. The results showed that SFR has excellent HOO$^\bullet$ antiradical activity with $k_{overall} = 4.75 \times 10^7 \, \text{M}^{-1} \, \text{s}^{-1}$ in water at pH = 7.40 by the SET pathway of the anion state, and good/moderate HOO$^\bullet$ scavenging activity in lipid environment ($k_{overall} = 1.79 \times 10^4 \, \text{M}^{-1} \, \text{s}^{-1}$) by the FHT mechanism via the O3′(O4′)–H bonds. The hydroperoxyl antiradical activity of SFR is better than Trolox, ascorbic acid and resveratrol in the lipid medium. This activity of SFR is approximately 530 times faster than that of Trolox and relatively similar to ascorbic acid and resveratrol in the polar environment. Thus, SFR can be a useful natural antioxidant in physiological environments.

Data accessibility. All relevant necessary data to reproduce all results in the paper are within the main text, electronic supplementary material and the Dryad Digital Repository: https://doi.org/10.5061/dryad.t4b8gtj1z [51].
The data are provided in the electronic supplementary material [52].
Authors' contributions. N.T.H., D.T.N.H., D.P.H. and H.V.T. carried out the molecular laboratory work, participated in data analysis, carried out sequence alignments, participated in the design of the study and drafted the manuscript; L.P.H. carried out the statistical analyses and collected field data; A.M. and Q.V.V. conceived of the study, designed the study, coordinated the study and helped draft the manuscript. All authors gave final approval for publication.
Competing interests. We declare we have no competing interests
Funding. This research is funded by the Vietnamese Ministry of Education and Training under project number B2021-DNA-16 (Q.V.V.).

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
