## [Peer Review File · Royal Society Open Science]

Review History

RSOS-210626.R0 (Original submission)

Review form: Reviewer 1 (Anoop Ayyappan)

Is the manuscript scientifically sound in its present form?

No

Are the interpretations and conclusions justified by the results?

No

Is the language acceptable?

Yes

Do you have any ethical concerns with this paper?

No

Have you any concerns about statistical analyses in this paper?

No

Recommendation?

Major revision is needed (please make suggestions in comments)

Comments to the Author(s)

Page 2: ONOO- or ONOO•.?

Page 3 Introduction paragraph 2. Provide the citation for Jung and coworkers ... Change inhibition to inhibition

Page 4 line 11 and Page5-line2: Change Gibb free energy to Gibbs free energy

Page 6 Table 3 caption: kECK, change k_{ECK} (subscript), and change -1 to superscript in M⁻¹S⁻¹

Page 6 kinetic study paragraph 2 below the table: Provide ΔG^\ddagger value after O3-H bond, similar to O4'-H bond above.

Page 9: The next table should be Table 4. Add a space after kcal/mol). Also change to 'As shown in Table 4'.

Page 5: The positive value 1.1 of ΔG^0 of a reaction (Table2 ΔG^0 RAF value for C2 is 1.1) does not rule out the possibility of the reaction especially considering the accuracy of the DFT calculation. The kinetics of the reaction mainly depend on the activation barrier. This is further supported by the kinetic study in this article that RAF at C8 does not contribute to the rate despite ΔG^0 being -4.6 kcal/mol. Therefore, I don't find any reason to exclude that path based only on the ΔG^0 values. To check the consistency, the calculations have to be carried out with few other methods (other functionals or wave-function based methods). The RAF path at C2 also should be included in the kinetic study.

According to PA value authors determined that the O4'-H bond is likely to dissociate. But in Figure 3 they showed the dissociation of the O6-H bond.

Review form: Reviewer 2**Is the manuscript scientifically sound in its present form?**

Yes

Are the interpretations and conclusions justified by the results?

Yes

Is the language acceptable?

Yes

Do you have any ethical concerns with this paper?

No

Have you any concerns about statistical analyses in this paper?

No

Recommendation?

Accept with minor revision (please list in comments)

Comments to the Author(s)

In this manuscript, the authors report a theoretical investigation on the scavenging activity of sulfuretin natural compounds against the OOH radical. Both thermodynamic and kinetic aspects have been considered. The study has been performed considering different reaction mechanism and environments (gas-phase, lipid and aqueous). The used computational protocol is quite standard and previously used by different authors. The pKa of the compounds has been also determined.

The manuscript is of interest for the journal readers, data are well discussed and conclusions are consistent with the obtained data. The manuscript can be published after the consideration of the following minor points:

- in water phase the major part of the OOH radical is present in the dissociated form. Why the authors neglect this data;
- The captions of some tables must be implemented (i.e. in table 3 it is not clear the environment);
- The language need moderate revision;
- Some typos must be removed.

Decision letter (RSOS-210626.R0)

Dear Dr Vo:

Title: The Hydroperoxyl Radical Scavenging Activity of Sulfuretin: Insights from Theory
Manuscript ID: RSOS-210626

The editor assigned to your manuscript has now received comments from reviewers. We would like you to revise your paper in accordance with the referee and Subject Editor suggestions which can be found below (not including confidential reports to the Editor). Please note this decision does not guarantee eventual acceptance.

Please submit your revised paper before 18-Jun-2021. Please note that the revision deadline will expire at 00.00am on this date. If we do not hear from you within this time then it will be assumed that the paper has been withdrawn. In exceptional circumstances, extensions may be possible if agreed with the Editorial Office in advance. We do not allow multiple rounds of revision so we urge you to make every effort to fully address all of the comments at this stage. If deemed necessary by the Editors, your manuscript will be sent back to one or more of the original reviewers for assessment. If the original reviewers are not available we may invite new reviewers.

On behalf of the Subject Editor Professor Anthony Stace and the Associate Editor Dr Debashree Ghosh.

RSC Associate Editor:

Comments to the Author:

The authors should revise their manuscript to incorporate the comments/ suggested changes by the referees and provide a point-by-point reply to the comments before the manuscript can be accepted.

RSC Subject Editor:

Comments to the Author:

(There are no comments.)

Reviewers' Comments to Author:

Reviewer: 1

Comments to the Author(s)

Page 2: ONOO- or ONOO•.?

Page 3 Introduction paragraph 2. Provide the citation for Jung and coworkers ... Change inhibition to inhibition

Page 4 line 11 and Page5-line2: Change Gibb free energy to Gibbs free energy

Page 6 Table 3 caption: k_{ECK}, change k_{ECK} (subscript), and change -1 to superscript in M⁻¹S⁻¹

Page 6 kinetic study paragraph 2 below the table: Provide ΔG^\ddagger value after O3-H bond, similar to O4'-H bond above.

Page 9: The next table should be Table 4. Add a space after kcal/mol). Also change to 'As shown in Table 4'.

Page 5: The positive value 1.1 of ΔG^0 of a reaction (Table2 ΔG^0 RAF value for C2 is 1.1) does not rule out the possibility of the reaction especially considering the accuracy of the DFT calculation. The kinetics of the reaction mainly depend on the activation barrier. This is further supported by the kinetic study in this article that RAF at C8 does not contribute to the rate despite ΔG^0 being -4.6 kcal/mol. Therefore, I don't find any reason to exclude that path based only on the ΔG^0 values. To check the consistency, the calculations have to be carried out with few other methods (other functionals or wave-function based methods). The RAF path at C2 also should be included in the kinetic study.

According to PA value authors determined that the O4'-H bond is likely to dissociate. But in Figure 3 they showed the dissociation of the O6-H bond.

Reviewer: 2

Comments to the Author(s)

In this manuscript, the authors report a theoretical investigation on the scavenging activity of sulfuretin natural compounds against the OOH radical. Both thermodynamic and kinetic aspects have been considered. The study has been performed considering different reaction mechanism and environments (gas-phase, lipid and aqueous). The used computational protocol is quite standard and previously used by different authors. The pKa of the compounds has been also determined.

The manuscript is of interest for the journal readers, data are well discussed and conclusions are consistent with the obtained data. The manuscript can be published after the consideration of the following minor points:

- in water phase the major part of the OOH radical is present in the dissociated form. Why the authors neglect this data;
- The captions of some tables must be implemented (i.e. in table 3 it is not clear the environment);
- The language need moderate revision;
- Some typos must be removed.

Author's Response to Decision Letter for (RSOS-210626.R0)

See Appendix A.

Decision letter (RSOS-210626.R1)

Dear Dr Vo:

Title: The Hydroperoxyl Radical Scavenging Activity of Sulfuretin: Insights from Theory
Manuscript ID: RSOS-210626.R1

It is a pleasure to accept your manuscript in its current form for publication in Royal Society Open Science. The chemistry content of Royal Society Open Science is published in collaboration with the Royal Society of Chemistry.

On behalf of the Subject Editor Professor Anthony Stace and the Associate Editor Dr Debashree Ghosh.

RSC Associate Editor

Comments to the Author:

The authors have addressed all the issues raised by the referees and I am happy to recommend that the manuscript be accepted.

Reviewer(s)' Comments to Author:

Appendix A

Dr Laura Smith
Publishing Editor, Journals

Royal Society of Chemistry
Thomas Graham House
Science Park, Milton Road
Cambridge, CB4 0WF

Dear Prof. Smith,

We have revised our manuscript meticulously following the reviewers' recommendations. Their comments allowed us to make several improvements to the manuscript. Please see the details in the response to reviewers file. Our responses are in blue and the changes are highlighted in red in the manuscript.

We believe that the manuscript is now ready for publishing.

Sincerely yours,

Quan Van Vo
The University of Danang - University of Technology and Education,
Danang 550000, Vietnam
Email: vvquan@ute.udn.vn
Danang, May 28, 2021

Reviewers' Comments to Author:

Reviewer: 1

Comments to the Author(s)

Page 2: ONOO- or ONOO•.?

Author reply: It is ONOO- , formed by the reaction of $O_2^{\bullet-}$ with NO free radical.

Page 3 Introduction paragraph 2. Provide the citation for Jung and coworkers ... Change inhibition to inhibition

Author reply: The reference has been updated. The typo has been corrected.

Page 4 line 11 and Page5-line2: Change Gibb free energy to Gibbs free energy

Author reply: The typo has been corrected.

Page 6 Table 3 caption: kECK, change k_{ECK} (subscript), and change -1 to superscript in $M^{-1}s^{-1}$

Author reply: The typos have been corrected.

Page 6 kinetic study paragraph 2 below the table: Provide ΔG^\ddagger value after O3-H bond, similar to O4'-H bond above.

Author reply: The ΔG^\ddagger value has been added.

Page 9: The next table should be Table 4. Add a space after kcal/mol). Also change to 'As shown in Table 4'.

Author reply: Done.

Page 5: The positive value 1.1 of ΔG^0 of a reaction (Table2 ΔG^0 RAF value for C2 is 1.1) does not rule out the possibility of the reaction especially considering the accuracy of the DFT calculation. The kinetics of the reaction mainly depend on the activation barrier. This is further supported by the kinetic study in this article that RAF at C8 does not contribute to the rate despite ΔG^0 being -4.6 kcal/mol. Therefore, I don't find any reason to exclude that path based only on the ΔG^0 values. To check the consistency, the calculations have to be carried out with few other methods (other functionals or wave-

function based methods). The RAF path at C2 also should be included in the kinetic study.

Author reply: The kinetic section has been updated with data for the C2 position following the RAF mechanism. However, the contribution of this reaction in the k_{overall} is minor.

We acknowledge the importance of choosing the right functional for calculating kinetics; however, this was luckily studied by several groups before us, and thus we could rely on their results. The Minnesota 06 functional is among the best methods to compute both thermodynamic and kinetic parameters with good accuracy that stand the comparison with results yielded by more complex functionals (i.e. G3(MP2)-RAD) or experimental data.¹⁻⁵ The M06-2X/6-311++G(d,p) level of theory has been widely used to evaluate the radical scavenging activity of organic compounds.^{1,6-11} The results obtained for RAF reactions here are also in good agreement with previous data in phenolic compounds.^{9,12,13} Thus, we are confident that the data is correct and reliable at the current state of the art of computational chemistry.

According to PA value authors determined that the O4'-H bond is likely to dissociate. But in Figure 3 they showed the dissociation of the O6-H bond.

Author reply: That was an error, it has been corrected.

Reviewer: 2

Comments to the Author(s)

In this manuscript, the authors report a theoretical investigation on the scavenging activity of sulfuretin natural compounds against the OOH radical. Both thermodynamic and kinetic aspects have been considered. The study has been performed considering different reaction mechanism and environments (gas-phase, lipid and aqueous). The used computational protocol is quite standard and previously used by different authors. The pKa of the compounds has been also determined.

The manuscript is of interest for the journal readers, data are well discussed and

conclusions are consistent with the obtained data. The manuscript can be published after the consideration of the following minor points:

-in water phase the major part of the OOH radical is present in the dissociated form. Why the authors neglect this data;

Author reply: We thank the reviewer for this suggestion. The HOO^\bullet could be deprotonated to form $\text{O}_2^{\bullet-}$ in water. Thus this process could be included in the rate constant by considering the molar fraction (f) of HOO^\bullet (at $\text{pH} = 7.40$, $f(\text{HOO}^\bullet) = 0.0025$). However, if we perform this adjustment for the reference antioxidants as well as the studied compounds, the values of rate constant will change to a similar degree; thus the trend is not affected only the absolute value. Therefore we did not include the deprotonation of HOO^\bullet . Since $\text{O}_2^{\bullet-}$ radical does not present in lipid media i.e., pentyl ethanoate, thus HOO^\bullet has been used as a model radical to evaluate the radical scavenging activity of antioxidants in the physiological environment.^{9,14-17}

- The captions of some tables must be implemented (i.e. in table 3 it is not clear the environment);

Author reply: The necessary information has been updated.

-The language need moderate revision;

-Some typos must be removed.

Author reply: The manuscript has been carefully revised, all of typos have been corrected.

References

1. M. Carreon-Gonzalez, A. Vivier-Bunge and J. R. Alvarez-Idaboy, *J. Comput. Chem.*, 2019, **40**, 2103-2110.
2. A. Galano and J. Raúl Alvarez-Idaboy, *Int. J. Quantum Chem.*, 2019, **119**, e25665.
3. Y. Zhao, N. E. Schultz and D. G. Truhlar, *J. Chem. Theory Comput.*, 2006, **2**, 364-382.
4. A. Galano and J. R. Alvarez-Idaboy, *J. Comput. Chem.*, 2014, **35**, 2019-2026.
5. Y. Zhao and D. G. Truhlar, *J. Phys. Chem. A*, 2008, **112**, 1095-1099.
6. H. Boulebd, I. A. Khodja, M. V. Bay, N. T. Hoa, A. Mechler and Q. V. Vo, *J. Phys. Chem. B*, 2020, **124**, 4123-4131.

7. H. Boulebd, A. Mechler, N. T. Hoa and Q. V. Vo, *New J. Chem.*, 2020, **44**, 9863-9869.
8. Q. V. Vo, N. T. Hoa, P. C. Nam, D. T. Quang and A. Mechler, *ACS Omega*, 2020, **5**, 24106–24110.
9. Q. V. Vo, P. C. Nam, M. Van Bay, N. M. Thong and A. Mechler, *RSC Adv.*, 2019, **9**, 42020-42028.
10. D. S. Dimić, D. A. Milenković, E. H. Avdović, Đ. J. Nakarada, J. M. D. Marković and Z. S. Marković, *Chemical Engineering Journal*, 2021, 130331.
11. D. A. Milenković, D. S. Dimić, E. H. Avdović, A. D. Amić, J. M. D. Marković and Z. S. Marković, *Chemical Engineering Journal*, 2020, **395**, 124971.
12. C. Iuga, J. R. I. Alvarez-Idaboy and N. Russo, *J. Org. Chem.*, 2012, **77**, 3868-3877.
13. M. Cordova-Gomez, A. Galano and J. R. Alvarez-Idaboy, *RSC Adv.*, 2013, **3**, 20209-20218.
14. A. Galano and J. Raúl Alvarez-Idaboy, *Int. J. Quantum Chem.*, 2019, **119**, e25665.
15. A. Galano and J. R. Alvarez-Idaboy, *J. Comput. Chem.*, 2013, **34**, 2430-2445.
16. Q. V. Vo, M. V. Bay, P. C. Nam, D. T. Quang, M. Flavel, N. T. Hoa and A. Mechler, *J. Org. Chem.*, 2020, **85**, 15514–15520.
17. Q. V. Vo, M. V. Bay, P. C. Nam and A. Mechler, *J. Phys. Chem. B*, 2019, **123**, 7777-7784.